# Clinical Benefits of Minimally Invasive Non-Surgical Periodontal Therapy as an Alternative of Conventional Non-Surgical Periodontal Therapy—A Pilot Study

**DOI:** 10.3390/ijerph19127456

**Published:** 2022-06-17

**Authors:** Wen-Chen Chung, Chiung-Fang Huang, Sheng-Wei Feng

**Affiliations:** 1School of Dentistry, College of Oral Medicine, Taipei Medical University, Taipei 110, Taiwan; chungology@gmail.com; 2Division of Periodontics, Department of Dentistry, Taipei Medical University Hospital, Taipei 110, Taiwan; 3Department of Dentistry, Hsinchu Cathay General Hospital, Hsinchu 300, Taiwan; 4Division of Family and Operative Dentistry, Department of Dentistry, Taipei Medical University Hospital, Taipei 110, Taiwan; 5Division of Prosthodontics, Department of Dentistry, Taipei Medical University Hospital, Taipei 110, Taiwan

**Keywords:** minimally invasive non-surgical periodontal therapy, root surface debridement, periodontitis

## Abstract

Minimally invasive procedures were introduced in periodontics, which could enhance clinical outcomes and reduce post-operative discomfort. However, minimally invasive non-surgical periodontal therapy (MINST) as an alternative modality of conventional non-surgical root surface debridement has not been clearly evaluated by randomized controlled clinical trial. The present study aimed to investigate clinical outcomes and patients’ comfort feedback of MINST compared to conventional non-surgical periodontal therapy (CNST). Patients with moderate to severe periodontitis were included. Nine out of ten patients were recruited and completed the post-treatment re-evaluation in this study. Randomized split-mouth design, CNST and MINST on each side, was performed. Clinical parameters, including periodontal probing depth (PD), gingival recession (REC), clinical attachment level (CAL), and gingival bleeding on probing (BOP), were recorded on baseline, 1 month and 3 months post-treatment. Non-parametric statistics were used for analysis. PD, REC, CAL, and BOP were improved after treatment in both CNST and MINST groups. Comfort feedback and gingival recession showed better outcomes in the MINST group than in the CNST group. No statistical significance of parameters was found between CNST and MINST. Within the limitations, minimally invasive non-surgical periodontal therapy could be an alternative modality of conventional non-surgical periodontal therapy. Further studies are required to establish clinical protocol and evidence of MINST.

## 1. Introduction

Periodontitis is a plaque-associated infection initiated by periodontal pathogen accumulation on surfaces of the teeth [1]. Non-surgical periodontal therapy has remained the fundamental treatment as an initial periodontal therapy for decades [2]. The primary goal of non-surgical periodontal therapy is to reduce periodontal inflammation and to re-establish periodontal health by closed-approach removal of supra- and sub-gingival biofilm and calculus [3,4]. Besides reducing local inflammation levels, the evidence also demonstrated that non-surgical periodontal therapy could improve systemic inflammation and glycemic level [5,6,7,8,9]. Scaling and root surface debridement play essential roles in non-surgical periodontal therapy [10,11].

Conventional non-surgical periodontal therapy (CNST) involves using power-driven and/or manual instrumentation to debride the root surfaces [2,12]. It was thought that bacteria by-products are infiltrated into the cementum, and meticulous root planning is necessary to remove diseased cementum by hand instruments for periodontal healing [13,14]. However, recent evidence has shown that intentional removal of root structures is not the prerequisite for successful periodontal treatment [15]. Gentle and light-touch root surface instrumentation to remove the plaque and calculus is sufficient [16,17].

Conversely, several studies have shown that residual calculus could be found after CNST, especially in pocket sites deeper than 5 mm [18,19,20]. In addition, blade face width of traditional periodontal curettes is unlikely to successfully treat the furcation area of multi-rooted teeth [21,22]. Regarding gingival recession after CNST, previous research studies have indicated that recession is inevitable and even more significant at sites with severe periodontal pockets [23,24].

Recently, minimally invasive surgical procedures have been introduced in periodontics [25,26,27]. Based on the concept of minimally invasive surgery, minimally invasive non-surgical periodontal therapy (MINST) has been introduced for minimizing tissue trauma and enhancing clinical outcomes by the adjunctive of magnification devices and specifically designed thin instruments during scaling and root surface debridement [28,29,30]. In contrast to the outcomes of soft/hard tissue trauma and residual calculus after CNST, MINST might be an alternative solution to overcome these limitations. Previous randomized controlled trials have shown that the reduction of periodontal intra-bony defect depth of MINST protocol is comparable to minimally invasive periodontal surgery [28]. However, thus far, there has been no previous study evaluating the efficacy and benefit of MINST as an alternative modality of CNST. This pilot study aimed to compare clinical parameters and patient comfort feedback between MINST and CNST as an initial periodontal therapy.

## 2. Materials and Methods

### 2.1. Study Design

This was a prospective, split-mouth single-blind randomized controlled pilot study conducted in Taipei Medical University Hospital from May 2020 until February 2022. Ethical approval was obtained from TMU-JIRB, Taipei Medical University (N202003155). The study was processed in accordance with the Helsinki Declaration. For the randomization process, a computer-generated list was used to define side and treatment modality. Only clinicians were aware of modality assignment.

### 2.2. Participant Selection

Participants were recruited from the dental department of the hospital. Potential patients were scheduled for clinical and radiographic examination to confirm the eligibility of enrollment. Inclusion criteria were as follows: systemically healthy; generalized moderate to severe (Stage II to IV) periodontitis, according to the American Academy of Periodontology (AAP) and the European Federation of Periodontology (EFP) periodontal classification system [31]; above age of 30; at least 5 retainable teeth in each quadrant. Exclusion criteria included smoking; systemic diseases that might affect periodontal health such as diabetes and autoimmune diseases [32]; previous periodontal treatment within 6 months before baseline examination; previous antibiotic regimen within 1 month before baseline examination; pregnancy.

### 2.3. Data Collection

All eligible participants were well-informed and signed the informed consent before baseline examination. Periodontal parameters included plaque index, periodontal probing depth (PD): the distance from the free gingival margin to the bottom of gingival sulcus, gingival recession depth (REC): the distance from free gingival margin to cementoenamel junction or restoration margin, clinical attachment level (CAL): the sum of PD and REC at individual site, bleeding on probing (BOP): recorded as bleeding site 10 s after probing, which were assessed at six sites per tooth at baseline, 1 month after treatment, and 3 months after treatment. The measurements of PD, REC, and CAL were recorded to the nearest 0.5 mm. The periodontal probe (PQ-W, Hu-Friedy, Chicago, IL, USA) was used for periodontal examination by a single experienced periodontist. Patients’ comfort feedback, which was evaluated by 10 cm horizontal visual analog scale (VAS) reporting their satisfaction of both modalities during treatment, were also collected. Self-reported dental habits including toothbrushing frequency and oral malodor were evaluated in a questionnaire at baseline and re-evaluation visits.

### 2.4. Split-Mouth and Intervention

The treatments were randomly divided into two groups for each patient. Teeth on one side of the mouth were treated by conventional non-surgical periodontal therapy; teeth on the other side were treated by minimally invasive non-surgical periodontal therapy. Participants received oral hygiene instruction including toothbrushing, interdental cleaning, and plaque control behavior reinforcement every visit; moreover, oral hygiene self-care regimens were asked to be followed at home. On the conventional non-surgical periodontal therapy side, a power-driven piezo-electric ultrasonic scaling device with scaling tips (Model P-10 and P-20, NSK, Tochigi, Japan) was used the majority of the time to debride subgingival biofilm and calculus, and manual standard Gracey curette instruments (Gracey curette 1/2; 11/12; 13/14, Hu-Friedy, Chicago, IL, USA) were used to refine residuals from root surfaces. Conversely, the treatment protocol of the minimally invasive non-surgical periodontal therapy side was followed in the previous study [33]. Briefly, the root surface of teeth was thoroughly debrided by a minimally invasive approach using specific thin and delicate piezo-electric scaling tips (Model P-26L, P-26R, P-40, NSK, Tochigi, Japan) and manual miniature periodontal curettes (Micro Mini-Five Gracey curette 1/2, 11/12, 13/14, Hu-Friedy, Chicago, IL, USA) for reducing periodontal tissue trauma. Adjunctive use of 4.0× magnification loupes was also necessary. Both modalities were performed under local anesthesia and were operated by a single experienced periodontist. Operation time was not restricted until the clinician was satisfied with the work. All patients were recalled 1 and 3 months after treatment for re-evaluation and professional supra-gingival plaque control.

### 2.5. Statistical Analysis

Periodontal parameters, including PD, REC, CAL, and BOP, and patients’ comfort feedback VAS score of inter-groups comparison were analyzed by Mann–Whitney U test; while intra-group comparisons between baseline and post-treatment were analyzed by Wilcoxon signed-rank test. All calculations were carried out using Graphpad Prism software, version 8.4.0 for Mac (Graphpad software Inc., San Diego, CA, USA). A *p* value < 0.05 was considered to indicate statistical significance.

## 3. Results

### 3.1. Sample and Demographics

As shown in Figure 1, an overview of patient recruitment and selection was illustrated. Ten patients were recruited and treated, and only one patient was excluded by absence on follow-ups. A total of nine participants completed the post-treatment re-evaluation. All patients were classified as generalized Stage III Grade B periodontitis cases. Baseline characteristics are summarized in Table 1. The majority of patients were female. The age of the included patients ranged from 33 to 62 years, and they presented poor oral hygiene behavior. The right side of six patients was treated by CNST; and the left side was treated by MINST. Conversely, the right side of three patients was treated by MINST, and the left side was treated by CNST. A total of 127 teeth in the CNST group and 123 teeth in MINST were included in this study.

### 3.2. Clinical Outcomes

Baseline parameters were similar between groups (*p* > 0.05) (Table 2). Average PD of CNST and MINST was 3.43 and 3.32 mm, respectively; average REC of CNST and MINST was 0.5 and 0.65 mm, respectively; average CAL of CNST and MINST was 3.88 and 3.92 mm, respectively; average BOP of CNST and MINST was 58% and 51.22%, respectively. At 1 and 3 months post-treatment re-evaluation visits, periodontal parameters, including PD, REC, CAL, and BOP, on both CNST and MINST groups were improved significantly (*p* < 0.05) (Figure 2) (Table 2). One month after treatment, average PD of CNST and MINST decreased 0.83 and 0.76 mm, respectively; average REC of CNST and MINST increased 0.35 and 0.33 mm, respectively; average CAL of CNST and MINST decreased 0.42 and 0.38 mm, respectively. Compared to baseline, three months after treatment, average PD of CNST and MINST decreased 0.92 and 0.83 mm, respectively; average REC of CNST and MINST increased 0.27 and 0.23 mm, respectively; average CAL of CNST and MINST decreased 0.6 and 0.55 mm, respectively. BOP of CNST decreased to 23.67% and 25.44% 1 and 3 months after treatment, respectively; and BOP of MINST decreased to 26.22% and 23.67% after treatment, respectively. Regarding inter-groups comparison, less gingival recession was shown 1 and 3 months after treatment on the MINST group than CNST group. However, there was no statistically significant difference of post-treatment improvement on PD, REC, CAL, and BOP between groups (*p* > 0.05) (Table 2).

Regarding differences of anterior teeth and posterior teeth, changes from baseline to 3 months post-treatment of anterior and posterior teeth between CNST and MINST were analyzed. Statistical significance was not found in changes of PD, REC, CAL, and BOP in different tooth position between CNST and MINST (*p* > 0.05) (Table 2). In comparison, between anterior teeth and posterior teeth in CNST, statistical significances were shown in changes of PD (*p* < 0.05) and CAL (*p* < 0.05); conversely, in comparison between anterior teeth and posterior teeth in MINST, statistical significances were shown in changes of PD (*p* < 0.05), CAL (*p* < 0.05), and BOP (*p* < 0.05). Posterior teeth showed better improvements in PD and CAL in both CNST and MINST. BOP was better improved in anterior teeth than in posterior teeth in the MINST group.

Time consumption of the two treatment modalities did not show a statistically significant difference (*p* > 0.05) (Table 2). Average time consumption for CNST was around 23.5 min, and for MINST was around 26 min. Regarding comfort feedback reported by patients, four patients responded no discomfort during the procedure of MINST; in contrast, only one patient felt no discomfort during treatment of CNST. VAS scores of CNST and MINST groups were 2.9 and 2.4 on average, respectively. However, the VAS results did not reveal statistical significance (*p* > 0.05) (Figure 3). Self-reported daily toothbrushing frequency was 2.22 ± 0.83 (mean ± SD) times at baseline. The toothbrushing frequency was increased to 3.0 ± 0.87 (mean ± SD) times and 3.22 ± 0.67 (mean ± SD) times at post-treatment 1 month and 3 months re-evaluation, respectively. Furthermore, five patients self-reported oral malodor at baseline; however, only tow patients at post-treatment 1 month re-evaluation visit and one patient at post-treatment 3 months re-evaluation visit still presented oral malodor.

## 4. Discussion

The use of minimally invasive therapy by adjunctive of magnification devices and fine instruments for treating periodontitis has been proposed recently, which might reduce tissue trauma, enhance healing potential, and decrease post-operative discomfort [34,35,36]. Minimally invasive periodontal surgical techniques were introduced to provide conservative procedures and wound healing stability [27]. Recently, based on the concept of minimally invasive periodontal surgery, MINST has been proposed as an alternative procedure of periodontal surgery in treating intra-bony defects [28]. The present study, to the best of our knowledge, is the first pilot study evaluating the differences between minimally invasive non-surgical periodontal therapy and conventional non-surgical periodontal therapy as an initial periodontal therapy.

A systematic review indicated that compared to shallow pocket sites (≤6 mm probing depth), unfavorable results of residual calculus, pocket depth reduction and clinical attachment level gain in deep pocket sites (>6 mm probing depth) after non-surgical periodontal therapy were found [37]. Residual deep periodontal pocket has been considered as a risk factor of further periodontal attachment loss and tooth loss [38]. Therefore, periodontal surgery might be indicated in those sites that could not be successfully treated by non-surgical therapy [39]. Graziani et al. conducted a meta-analysis of 647 subjects in 27 randomized clinical trials calculating the weighted means of 1.65 mm CAL gain, 2.8 mm PD reduction, and 1.26 mm REC increase 12 months after periodontal access surgery [40]. However, burdens of psychological, economical, and post-operative discomfort of patients after periodontal surgery should be considered [39]. A previous retrospective study evaluated 35 intra-bony defects treated by MINST; and the results showed that average PD and clinical attachment loss reduction were 3.12 and 2.18 mm, respectively, at intra-bony defect sites [33]. A systematic review analyzed clinical efficacy of MINST and minimally invasive periodontal surgical modalities in intra-bony defects [41]. The results found that MINST compared to surgical treatment showed the lowest probability as the best modality option for CAL gain in intra-bony defects. However, these research studies compared MINST to surgical therapies, and the data from these research studies were extracted from isolated deep defect sites. There is still a lack of evidence of differences between MINST and CNST. Therefore, our study compared two different non-surgical periodontal therapies, MINST and CNST, and evaluated average full-mouth periodontal parameters.

Apparent gingival recession is another inevitable outcome after conventional non-surgical periodontal therapy, especially in gingiva with significant inflammation [24,42]. This outcome is mainly due to subsidence of swelling and tissue shrinkage after periodontal treatment [23]. The systematic review has shown that REC increases 3 months after non-surgical periodontal treatment with or without systemic antibiotics, ranging from a median value of 0.14 to 0.3 mm in patients with chronic periodontitis [24]. In addition, in aggressive periodontitis cases, the median value of REC was 0.2 mm 3 months after non-surgical treatment [24]. Nibali et al. evaluated 35 consecutive intra-bony defects treated by MINST in a retrospective study. The results showed that the mean change of REC was 0.3 mm from baseline to 12 months after treatment [33]. Limited vision and inadequate instrumentation access during treatment might damage gingival tissue stability, impair the healing process and further aggravate these clinical outcomes [27,34]. In our study, gingival recession was increased one month after both treatments; however, slight gingival tissue rebound occurred at 3 months evaluation. Furthermore, less gingival recession after treatment was found in the MINST group than in the CNST group; however, this result was not statistically significant.

Manual instruments, such as standard Gracey curettes, are designed mainly with a relatively long and straight blade. When removing deposits on the root surface, it easily leaves some gaps without thorough debridement of the root surface [43]. Failure of total removal of dental calculus on treated root surfaces was found in several studies, especially in deep periodontal pockets [20,44]. Newly developed Gracey curettes with longer shanks and thinner and smaller blades may allow for better access to complicated root areas, facilitate contamination debridement, and minimize hard and soft tissue trauma versus standard Gracey curettes [45]. An earlier study compared the effectiveness of subgingival calculus removal in 35 non-molar teeth between standard Gracey curettes, newly designed longer shank and thinner blade Gracey curettes (After Five Gracey curette, Hu-Friedy, Chicago, IL, USA), and untreated control [44]. The results indicated a significant treatment effect in terms of percentage of residual calculus compared to untreated teeth. Moreover, the mesial tooth surfaces presented the least residual calculus. However, there is no significant difference in percentage of residual calculus between standard Gracey curettes and the newly designed one. In contrast, Acunzo et al. treated 109 teeth with either conventional standard Gracey curettes or with newly designed small-sized Mini-Five Gracey curettes (Hu-Friedy, Chicago, IL, USA) [46]. They found that the use of Mini-Five Gracey curettes resulted in greater PD reduction and lower gingival recession. Regarding power-driven devices, a recent randomized clinical trial compared three different sizes of piezoelectric ultrasonic scaler tips for in vitro and clinical analyses [47]. The results showed that using a slim-designed scaler tip caused less tooth substance loss and less pain experience for patients than a conventional wide scaler tip. These studies supported that treating root surfaces with slim and small instruments might facilitate better outcomes than conventional instruments. In our study, teeth of the MINST groups were treated by Micro Mini-Five Gracey curettes, which were designed to have an elongated terminal shank and 50% shorter blade than standard Gracey curettes. In addition, the thickness of the Micro Mini-Five Gracey curettes is 20% thinner than the Mini-Five Gracey curettes. The results in the present study also revealed a similar trend of more favorable clinical outcomes from using small-diameter instruments versus conventional ones.

Poor visualization could lead to unsatisfied operation work, especially in dentistry, which faces small and complicated anatomical structures. Magnification devices are widely used in medical and dental practice currently [48,49]. Magnification equipment, either surgical loupes or microscopes, can facilitate precise visualization clinically, including clinicians’ working posture [50]. Eichenberger et al. evaluated the precision of tooth preparation of 16 dentists aged 26 to 67 years using either loupe, microscope, or without any kind of magnification device [51]. They found that the precision of preparation was significantly higher when using microscope, followed by loupe. The precision was the lowest in preparation without any magnification device. A previous comparative study demonstrated that manual scaling and root planning treatment with magnification loupes could result in less damage to the tooth structure and in less remaining calculus [52]. However, a recent study compared a total of 30 patients in supragingival debridement with and without using 2.5× magnification loupes [53]. They concluded that the use of 2.5× magnification loupes did not improve clinical and patient-centered outcomes significantly. Surgical loupes were used for assisting in debridement work in the MINST group in the present study. Although there was no significant difference between the two treatment groups, we found less gingival shrinkage in the MINST group.

Handling different minimally invasive instruments and gently performing minimally invasive treatment might increase operation time compared to conventional treatment [54]. A recent systematic review and meta-analysis including 19 studies analyzed the difference in efficacy between microsurgery and macrosurgery for gingival recession treatment [55]. Based on their findings, they suggested that the clinical outcomes of root coverage were better improved by microsurgery than by macrosurgery. The complete root coverage was significantly higher when using microscope, followed by loupes, versus macrosugery control. This is possibly owed to the minimally invasive approach and precision technique when using magnification devices. However, treatment time was increased for microsurgical procedures versus microsurgery procedures. Similar results were found in the present study. Our study showed that MINST could take longer chair time than CNST. This result may be attributed to spending time changing instruments and to the delicate debridement of the root surface. Conversely, according to the previous randomized clinical trial, MINST presented an advantage of saving chair time compared to minimally invasive surgical procedures [28].

Conventional periodontal treatment is usually related to discomfort experienced by patients [56,57,58]. A recent systematic review included 13 studies to investigate the relationship between CNST and patient-based outcomes [59]. Based on the results, physical pain was a common finding reported by patients after treatment. Matthews et al. quantified patient perception of various periodontal non-surgical and surgical techniques by VAS [57]. They found that patients’ perception during CNST was not satisfactory. Moreover, in contrast to non-surgical procedures, surgery produced significantly higher post-operative discomfort. Mei et al. assessed 253 patients’ perceptions following 330 periodontal or implant surgeries [60]. The majority of patients experienced mild post-operative pain; 70.3% of patients perceived mild pain, 25.5% perceived moderate pain, and 4.2% perceived severe pain. The median of self-reported duration of pain was 2 days. The highest post-surgical pain was found in subjects receiving advanced implant surgery; and the lowest was in open flap debridement. In our study, patients’ comfort feedback after both modalities of non-surgical periodontal therapy seemed to show promising results. VAS score was less in the MINST group than in CNST; however, this result did not show statistical significance. Furthermore, the number of patients reporting no discomfort feedback were higher in the MINST group than in the CNST group, although the result was not statistically significant. In accordance with the present results, a previous study also demonstrated satisfactory patient perception during the MINST procedure [28].

Since the study was designed as a pilot study, the possible weakness of the sample size could lead to reducing the power of the study and to non-significant outcomes. Therefore, non-parametric statistics were appropriate for this limitation. The small sample size could also present possible study bias, e.g., gender distribution. Sexual dimorphism may exist in the prevalence of periodontitis. A systematic review representing 50,604 subjects from 12 population surveys found that sex exhibited significant association with periodontitis prevalence. Men appeared at higher risk for destructive periodontitis than women [61]. Future studies could clearly calculate an appropriate sample size based on the results of the present study. Second, the re-evaluation interval in the present study was 1 and 3 months after treatment. Extending the follow-up interval to 6 or 12 months after treatment could reveal long-term outcomes of MINST. Third, the absence of subgingival microbiological and inflammatory biomarker evidence in our study could reveal changes in periodontal pathogens and inflammatory cytokines after different modalities. This could be analyzed in future studies to evaluate the microbiological change and difference of the two modalities. However, based on this randomized split-mouth design pilot study, the trend of encouraging results after MINST is obvious. Further studies are required to establish clinical protocol and evidence of MINST.

## 5. Conclusions

Minimally invasive non-surgical periodontal therapy could be an alternative modality of conventional non-surgical periodontal therapy. Clinical outcomes of MINST were promising. Both MINST and CNST provided significant clinical results.

## Figures and Tables

**Figure 1 ijerph-19-07456-f001:**
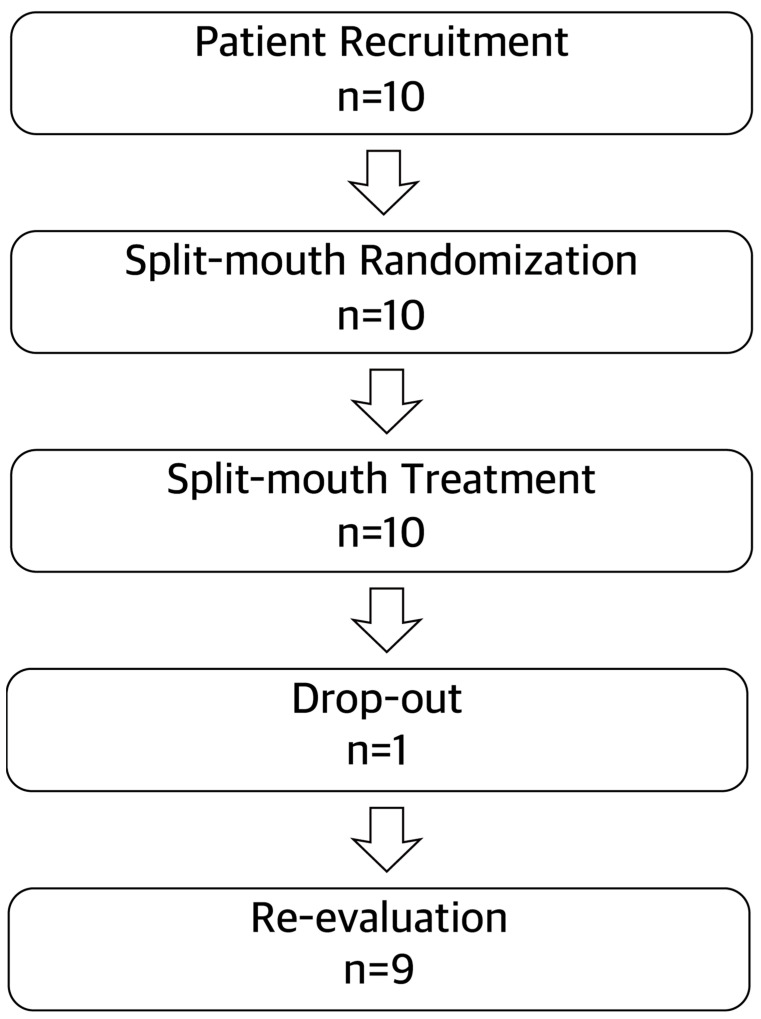
Flowchart.

**Figure 2 ijerph-19-07456-f002:**
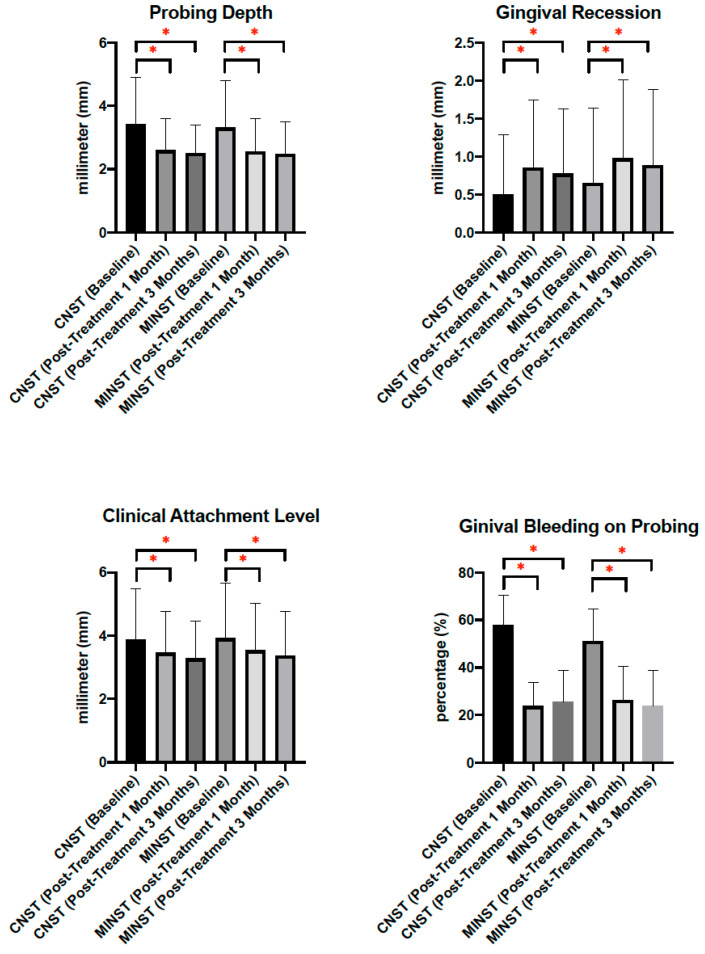
Periodontal parameters. Mann–Whitney U test was used for inter-group comparison, and Wilcoxon signed-rank test was used for intra-group comparison. *: *p* value < 0.05.

**Figure 3 ijerph-19-07456-f003:**
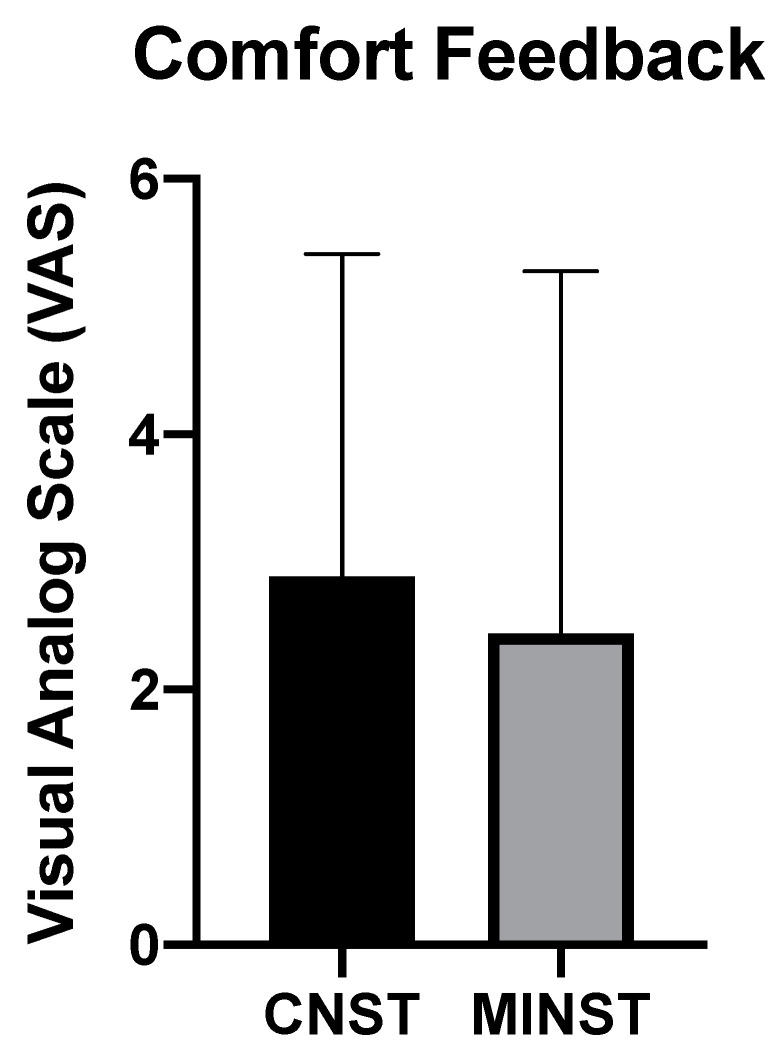
Patients’ comfort feedback. Analyzed by Mann–Whitney U test. Statistical significance set as *p* value < 0.05.

**Table 1 ijerph-19-07456-t001:** Baseline characteristics.

	Patients (n = 9)
Gender	
Male (n)	1
Female (n)	8
Age (mean, range)	47.3, 33–62
Number of teeth	250
Number of sites with PD < 5 mm	1185
Full mouth bleeding score (mean ± SD)	54.61 ± 13.15
Full mouth plaque score (mean ± SD)	94.73 ± 6.28

**Table 2 ijerph-19-07456-t002:** Clinical outcomes of CNST and MINST.

	CNST (n = 9)	MINST (n = 9)	Significance
Periodontal probing depth (mm) (mean ± SD)			
Baseline	3.43 ± 1.47	3.32 ± 1.49	n.s.
Post-treatment 1 month	2.60 ± 0.99	2.56 ± 1.05	n.s.
Post-treatment 3 months	2.51 ± 0.90	2.49 ± 1.01	n.s.
Gingival recession depth (mm) (mean ± SD)			
Baseline	0.50 ± 0.79	0.65 ± 0.99	n.s.
Post-treatment 1 month	0.85 ± 0.9	0.98 ± 1.03	n.s.
Post-treatment 3 months	0.77 ± 0.85	0.88 ± 1.00	n.s.
Clinical attachment level (mm) (mean ± SD)			
Baseline	3.88 ± 1.60	3.92 ± 1.74	n.s.
Post-treatment 1 month	3.46 ± 1.31	3.54 ± 1.48	n.s.
Post-treatment 3 months	3.28 ± 1.19	3.37 ± 1.39	n.s.
Gingival bleeding on probing (%) (mean ± SD)			
Baseline	58 ± 12.66	51.22 ± 13.47	n.s.
Post-treatment 1 month	23.67 ± 10.19	26.22 ± 14.23	n.s.
Post-treatment 3 months	25.44 ± 13.24	23.67 ± 15.03	n.s.
	Difference between CNST and MINST (from baseline to 3 months post-treatment)		
Changes of periodontal probing depth (mm) (mean ± SEM)			
Anterior teeth	0.05 ± 0.06		n.s.
Posterior teeth	0.11 ± 0.10		n.s.
Changes of gingival recession depth (mm) (mean ± SEM)			
Anterior teeth	0.02 ± 0.06		n.s.
Posterior teeth	−0.08 ± 0.05		n.s.
Changes of clinical attachment level (mm) (mean ± SEM)			
Anterior teeth	0.17 ± 0.08		n.s.
Posterior teeth	−0.03 ± 0.11		n.s.
Changes of bleeding on probing (%) (mean ± SEM)			
Anterior teeth	−0.03 ± 0.04		n.s.
Posterior teeth	0.06 ± 0.04		n.s.
Time consumption (second)	1412 ± 609.8	1559 ± 516.1	n.s.

Wilcoxon signed-rank test was used for intra-group comparison. Statistical significance set as *p* value < 0.05. n.s.: no significant difference.

## Data Availability

Data will be available from the authors upon reasonable request.

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
