# Peer review of "Clinical Benefits of Minimally Invasive Non-Surgical Periodontal Therapy as an Alternative of Conventional Non-Surgical Periodontal Therapy—A Pilot Study"

_ijerph, 2022, doi:10.3390/ijerph19127456_

Round 1

Reviewer 1 Report

This is a potentially interesting subject however the article has major flaws.

The use of English must be revised and corrected.

The abstract must be totally redone.

Please see the enclosed pdf for further details.

Author Response

Abstract

ANS: we followed the suggestions to revise our manuscript.

Introduction

ANS: we followed the suggestions to revise our manuscript.

Materials and Methods

ANS: we did require patients to perform oral hygiene self-care regimens.

This study is a randomized controlled split mouth design study which all patients were evaluated and operated by same clinician. Bias was limited as small as possible. However, bias analysis was not included in this study.

Discussion

ANS: we followed the suggestions to revise our manuscript.

Conclusion

ANS: we followed the suggestions to revise our manuscript.

Reviewer 2 Report

Dear Authors thanks for giving me the opportunity to review your work.

The sample size seems very little even for a pilot study.  Please justify from a statistical stand point how your results can be useful in planning a future study with a bigger sample size.

Usually a paired t-test using the difference between the baseline and post-treatment measurements and full data is the test that performed most appropriately in the different clinical scenarios when it is assumed that the treatment had a constant effect across teeth and subjects, and a paired t-test using the difference between the baseline and post-treatment measurements and collapsed data is the test that performed most appropriately in the different clinical scenarios when it was assumed that the treatment had a constant effect across teeth but not across subjects.

Please explain and discuss the rational of your decision in order of data analysis.

Differential gene regulation, particularly in sex steroid-responsive genes, may contribute to a sexual dimorphism in susceptibility to destructive periodontal disease or response to periodontal therapy. Please address this topic in your discussion due to the sex etherogenity of the study sample

Author Response

The sample size seems very little even for a pilot study.  Please justify from a statistical stand point how your results can be useful in planning a future study with a bigger sample size.

ANS: This is a randomized split-mouth design pilot study, the patient sample number is small, but 250 teeth entered the experiment (CNST:127 teeth; MINST:123 teeth). We can get a preliminary understanding of the trend from this prodromal study.

We added the description. “Since the study was designed as a pilot study, the possible weakness of sample size would lead to reducing power of the study, and non-significant outcomes. Therefore, non-parametric statistics were appropriate for this limitation. Small sample size could also present possible study bias, for e.g., gender distribution. Sexual dimorphism may exist in the prevalence of periodontitis. A systematic review represented 50,604 subjects from 12 population surveys. They found that sex exhibited significant association with periodontitis prevalence. Men appeared at higher risk for destructive periodontitis than women [61]. Future studies could clearly calculate the appropriate sample size based on the results of present study.”

Usually a paired t-test using the difference between the baseline and post-treatment measurements and full data is the test that performed most appropriately in the different clinical scenarios when it is assumed that the treatment had a constant effect across teeth and subjects, and a paired t-test using the difference between the baseline and post-treatment measurements and collapsed data is the test that performed most appropriately in the different clinical scenarios when it was assumed that the treatment had a constant effect across teeth but not across subjects.

Please explain and discuss the rational of your decision in order of data analysis.

ANS: Due to small sample size, we used non-parametric Wilcoxon signed-rank test for evaluate the intra-group differences between pre-treatment and post-treatment.

Differential gene regulation, particularly in sex steroid-responsive genes, may contribute to a sexual dimorphism in susceptibility to destructive periodontal disease or response to periodontal therapy. Please address this topic in your discussion due to the sex etherogenity of the study sample

ANS: we added the description. “Small sample size could also present possible study bias, for e.g., gender distribution. Sexual dimorphism may exist in the prevalence of periodontitis. A systematic review represented 50,604 subjects from 12 population surveys. They found that sex exhibited significant association with periodontitis prevalence. Men appeared at higher risk for destructive periodontitis than women [61].”

Reviewer 3 Report

Manuscript ID: ijerph-1677169

„Clinical Benefits of Minimally Invasive Non-Surgical Periodontal Therapy as an Alternative of Conventional Non-Surgical Periodontal Therapy. A Pilot Study” by Chung et al

The manuscript describes a prospective single blinded randomized controlled study performed on 9 patients (1 male, 8female, mean age, range 47.3, 33-62) suffering from periodontitis (stage II to IV) using a split-mouth design. The aim was to compare minimally invasive non-surgical periodontal therapy (MINST) with conventional non-surgical periodontal therapy (CNST) after 1 and 3 month(s) by assessing PD, REC, CAL and BOP. Statistical analyses were performed using non-parametric tests. Essentially the authors found no significant differences regarding PD, REC, CAL and BOP between MINST and CNST treated patients. 

MINST was introduced to obtain extensive subgingival debridement with minimal tissue trauma. Only few studies already investigated the effectiveness of the concept. The present study could therefore add to the knowledge on MINST.

However, the cohort examined in the present study is small and furthermore it is unclear (randomization?) why only 3 (of 9) patients were treated by MINST. Of course, this clearly limits the generalizability of the study.

At least the authors should add tables indicating detailed patients’ characteristics (No. of teeth, No. PPD<4mm, FMBS, FMPS). Results should also be presented as changes and anterior teeth should be compared to posterior teeth (PD, REC, CAL and BOP).

The discussion section should focus in the actual results and can thus be shortened.

The manuscript needs language editing.

Recommendation: Accept with major revision         

Author Response

MINST was introduced to obtain extensive subgingival debridement with minimal tissue trauma. Only few studies already investigated the effectiveness of the concept. The present study could therefore add to the knowledge on MINST.

ANS: The current evidence of MINST is still scarce. We added the following sentence in introduction section. “In contrast to the outcomes of soft/hard tissue trauma and residual calculus after CNST, MINST might be an alternative solution to overcome these limitations.” And we demonstrated MINST concept and citations on discussion section.

However, the cohort examined in the present study is small and furthermore it is unclear (randomization?) why only 3 (of 9) patients were treated by MINST. Of course, this clearly limits the generalizability of the study.

ANS: the study was a split-mouth design, which random divide each patient’s mouth to left or right side, and each side was treated by CNST or MINST. Therefore, 9 patients were all treated by CNST and MINST.

At least the authors should add tables indicating detailed patients’ characteristics (No. of teeth, No. PPD<4mm, FMBS, FMPS). Results should also be presented as changes and anterior teeth should be compared to posterior teeth (PD, REC, CAL and BOP).

ANS: we added the patients’ characteristics in Table 1. The changes were demonstrated in Figure 2. Changes of anterior teeth and posterior teeth were shown in Table 2.

We added the description in results section. “Regarding differences of anterior teeth and posterior teeth, changes from baseline to 3-months post-treatment of anterior and posterior teeth between CNST and MINST were analyzed. Statistical significance was not found in changes of PD, REC, CAL, and BOP in different tooth position between CNST and MINST (p>0.05) (Table 2). In comparison between anterior teeth and posterior teeth in CNST, statistical significances were shown in changes of PD (p<0.05), CAL (p<0.05); on the other hand, in comparison between anterior teeth and posterior teeth in MINST, statistical significances were shown in changes of PD (p<0.05), CAL (p<0.05), and BOP (p<0.05).”

The discussion section should focus in the actual results and can thus be shortened.

ANS: we have modified our discussion section.

Reviewer 4 Report

Abstract

Well structured and informative. No remarks

Introduction 

Please in a few sentences describe the main characteristics of MINST and main differences of MINST vs. conventional NST. 

Materials and methods

No remarks

Results

No remarks

Discussion

Page 8, line 223 the authors state: “ A systematic review analyzed clinical efficacy of MINST and minimally invasive periodontal surgical modalities in intra-bony defects [41]. The results found that MINST comparing to surgical treatment was the lowest probability to be the best modality option for CAL gain in intrabony defects. However, the data from these researches were extracted from isolated deep defect sites. Based on the results of our study, average full-mouth PD and CAL improvement after MINST are statistically significant.” - The data from the systematic review compared different minimally invasive surgical techniques to minimally invasive non-surgical and concluded that the surgical option was superior to non-surgical option n for CAL gain in intra-bony defects. This study compared two non-surgical options. Therefore, the data are not comparable. 

Page 9, line 243 - authors state that “...less gingival recession after treatment was found in MINST group than in CNST group.” - it must be emphasised that the difference was not statistically significant 

Page 10, line 318 - authors state that “VAS score  was less in MINST group than in CNST.”  it must be emphasised that the difference was not statistically significant 

“Furthermore, number of patients reported no discomfort feedback were higher in MINST group than in CNST group.” - was this statistically significant? Please state. 

Main limitation of the study i.e. small number of participants is mentioned with only one sentence. Please elaborate more on that. 

The conclusion

Page 11, Line 336 “it seemed that minimally invasive non-surgical periodontal therapy resulted in less gingival recession after treatment” is misleading as the results were not statistically significant. Please rephrase accordingly. 

Author Response

Introduction 

Please in a few sentences describe the main characteristics of MINST and main differences of MINST vs. conventional NST. 

ANS: we added the following sentence in introduction section. “In contrast to the outcomes of soft/hard tissue trauma and residual calculus after CNST, MINST might be an alternative solution to overcome these limitations.”

Discussion

Page 8, line 223 the authors state: “ A systematic review analyzed clinical efficacy of MINST and minimally invasive periodontal surgical modalities in intra-bony defects [41]. The results found that MINST comparing to surgical treatment was the lowest probability to be the best modality option for CAL gain in intrabony defects. However, the data from these researches were extracted from isolated deep defect sites. Based on the results of our study, average full-mouth PD and CAL improvement after MINST are statistically significant.” - The data from the systematic review compared different minimally invasive surgical techniques to minimally invasive non-surgical and concluded that the surgical option was superior to non-surgical option n for CAL gain in intra-bony defects. This study compared two non-surgical options. Therefore, the data are not comparable. 

ANS: We followed the suggestion to modified our manuscript.

Page 9, line 243 - authors state that “...less gingival recession after treatment was found in MINST group than in CNST group.” - it must be emphasised that the difference was not statistically significant 

 ANS: We followed the suggestion to modified our manuscript.

Page 10, line 318 - authors state that “VAS score  was less in MINST group than in CNST.”  it must be emphasised that the difference was not statistically significant 

ANS: We followed the suggestion to modified our manuscript.

“Furthermore, number of patients reported no discomfort feedback were higher in MINST group than in CNST group.” - was this statistically significant? Please state. 

ANS: We followed the suggestion to modified our manuscript.

Main limitation of the study i.e. small number of participants is mentioned with only one sentence. Please elaborate more on that. 

 ANS: We followed the suggestion to modified our manuscript.

The conclusion

Page 11, Line 336 “it seemed that minimally invasive non-surgical periodontal therapy resulted in less gingival recession after treatment” is misleading as the results were not statistically significant. Please rephrase accordingly. 

ANS: We followed the suggestion to modified our manuscript.

Round 2

Reviewer 2 Report

Dear Authors 

Within the limitations of the study the paper has improved

Author Response

Thank you for kindly suggestion.

Reviewer 3 Report

While the data were supplemented as suggested, the changes in the text were made only half-heartedly; in particular, the discussion is still poorly focused and much too long compared to the significance (also due to the small number of cases) of the study.

However, I would like to leave the decision to the editor.

I can accept the publication with the data now offered.

Author Response

Thank you for kindly suggestion.

This manuscript is a resubmission of an earlier submission. The following is a list of the peer review reports and author responses from that submission.